# Prognostic Value of Redox Status Biomarkers in Patients Presenting with STEMI or Non-STEMI: A Prospective Case-Control Clinical Study

**DOI:** 10.3390/jpm13071050

**Published:** 2023-06-26

**Authors:** Zorica Savovic, Bozidar Pindovic, Maja Nikolic, Ivan Simic, Goran Davidovic, Vladimir Ignjatovic, Jelena Vuckovic, Nenad Zornic, Tamara Nikolic Turnic, Vladimir Zivkovic, Ivan Srejovic, Sergej Bolevich, Vladimir Jakovljevic, Violeta Iric Cupic

**Affiliations:** 1Department of Internal Medicine, Faculty of Medical Sciences, University of Kragujevac, 34000 Kragujevac, Serbia; zoricasavo@gmail.com (Z.S.);; 2Department of Pharmacy, Faculty of Medical Sciences, University of Kragujevac, 34000 Kragujevac, Serbia; 3Department of Physiology, Faculty of Medical Sciences, University of Kragujevac, 34000 Kragujevac, Serbia; 4Department of Cardiology, University Clinical Center Kragujevac, 34000 Kragujevac, Serbia; 5Department of Surgery, Faculty of Medical Sciences, University of Kragujevac, 34000 Kragujevac, Serbia; 6N.A. Semashko Public Health and Healthcare Department, F. F. Erismann Institute of Public Health, I. M. Sechenov First Moscow State Medical University (Sechenov University), 119435 Moscow, Russia; 7Center of Excellence for Redox Balance Research in Cardiovascular and Metabolic Disorders, 34000 Kragujevac, Serbia; 8Department of Clinical Pharmacology, I. M. Sechenov First Moscow State Medical University (Sechenov University), 119435 Moscow, Russia; 9Department of Human Pathology, 1st Moscow State Medical, University I. M. Sechenov, 119991 Moscow, Russia

**Keywords:** acute coronary syndrome, myocardial infarction, myocardial reperfusion, non-ST-elevated myocardial infarction, biomarkers of oxidative stress, antioxidant capacity

## Abstract

(1) Background: The aim of our study was to determine the role of oxidative stress (OS) during early evaluation of acute ST-elevated myocardial infarction (STEMI) and non-ST-elevated myocardial infarction (NSTEMI) patients in order to define the role of redox balance in profiling the development of myocardial infarction (MI). (2) Methods: This prospective observational case-control study included 40 consecutive STEMI and 39 NSTEMI patients hospitalized in the coronary care unit of the cardiology clinic at the Kragujevac Clinical Center, Serbia, between 1 January 2016 and 1 January 2017. Blood samples were collected from all patients for measuring cardio-specific enzymes at admission and 12 h after admission to evaluate systemic oxidative stress biomarkers and the activity of antioxidant enzymes. (3) Results: In this study, participants were predominately female (52%), with a mean age of 56.17 ± 1.22 years old in the STEMI group and 69.17 ± 3.65 in the non-STEMI group. According to the Killip classification, the majority of patients (>50%) were at the second and third level. We confirmed the elevation of superoxide anion radicals in the non-STEMI group 6 h after admission in comparison with the STEMI and CTRL groups, but levels had decreased 12 h after admission. Levels of hydrogen peroxide were statistically significantly increased in the NSTEMI group. A positive correlation of superoxide anion radicals and levels of troponin I at admission was observed (r = 0.955; *p* = 0.045), as well as an inverse correlation between reduced glutathione and levels of NT-pBNP measured 6 h after admission (r = −0.973; *p* = 0.027). (4) Conclusions: We confirmed that superoxide anion radicals and reduced glutathione observed together with hs-troponin I at admission and NT-pBNP during hospital treatment could be predictors of ST evolution.

## 1. Introduction

Acute coronary syndrome (ACS) encompasses several clinical conditions, including unstable angina pectoris (USAP), ST-segment elevation myocardial infarction (STEMI), and non-ST-segment elevation myocardial infarction (NSTEMI) [1]. Despite a significant progress in understanding of pathophysiological mechanisms underlining ACS, as well as diagnostic and therapeutic algorithms, ACS still remains one of the leading causes of death in both developed and developing countries [2]. In 2015, the American Heart Association reported that approximately 370,000 people die from myocardial infarction (MI) in the United States annually, with one person experiencing MI every 42 s [3].

All forms of ACS share a common pathophysiological substrate—an unstable atherosclerotic plaque complicated by occlusive or non-occlusive thrombosis. In STEMI patients, a typical pathophysiological substrate is a complete coronary artery occlusion, followed by transmural ischemia and necrosis, while NSTEMI patients most frequently experience flow-limiting stenosis due to transient or incomplete coronary artery occlusion, leading to subendocardial ischemia and necrosis [4]. Both STEMI and NSTEMI share a similar clinical presentation in relation to signs and symptoms [5]. According to the observational study by Montalescot et al., intrahospital and long-term prognosis for both STEMI and NSTEMI patients was very similar, despite their pathophysiological differences [6].

A growing body of evidence indicates that oxidative stress (OS) and reactive oxygen species (ROS) have a crucial role in the pathogenesis of atherosclerosis and its progression toward ACS [7]. OS is generally referred to as a disproportion between reactive oxygen species (ROS) production and antioxidant protective capacity [8]. ROS occur as a physiological by-product of cellular metabolism in aerobic organisms, in the form of free radicals and non-radical forms. In low concentrations, ROS act as signaling molecules involved in various physiological processes [8]. However, in the case of their excessive production or insufficient antioxidant protection, ROS may initiate a cascade of aggravating events, such as the oxidation of macromolecules, deoxyribonucleic acid (DNA) damage, lipid peroxidation, damage to cell membranes, and apoptosis [9,10]. In the pathophysiological setting of coronary artery disease (CAD), ROS contribute to the formation and progression of atherosclerosis, along with plaque instability and rupture [11,12]. In addition, ROS are involved in the mechanism of myocardial reperfusion injury and further cardiomyocyte damage upon revascularization [13,14,15]. Therefore, some authors have suggested that ROS levels could have a predictive significance in the assessment of myocardial recovery after MI [16,17,18].

Various studies have highlighted the impaired redox status in patients with stable CAD and ACS in comparison to healthy individuals [19]. Furthermore, patients with unstable angina pectoris (USAP) had a higher ROS production than those with stable angina pectoris (SAP) with the same degree of angiographic narrowing of the coronary arteries, emphasizing the importance of OS in CAD, which is potentially greater than the degree of coronary stenosis alone [20].

Assessment of OS biomarkers could thus improve further pathophysiological understanding of ACS, modify risk stratification and treatment, and ultimately improve the clinical course of those patients. Therefore, the aim of our study was to determine the roles played by OS during the early evaluation of acute STEMI and NSTEMI patients in order to define the role of redox balance in profiling the development of MI. In addition, we wanted to obtain information about the possible predictors of developing STEMI or non-STEMI.

## 2. Materials and Methods

### 2.1. Design of Study

This prospective observational case-control study included 40 consecutive STEMI (STEMI group) and 39 NSTEMI patients (non-STEMI group) hospitalized in the coronary care unit of the cardiology clinic at the Kragujevac Clinical Center, Serbia, between 1 January 2016 and 1 January 2017. The control group of patients (CTRL) was comprised of 20 healthy volunteers without a history of CAD, matched with the other groups.

### 2.2. Patients

Patients with ACS were divided into two groups: patients with occlusive lesion and reperfusion (STEMI infarction) and patients with non-occlusive lesion (NSTEMI infarction). ST-segment elevation myocardial infarction (STEMI) and non-ST-segment elevation myocardial infarction (NSTEMI) designations were based on the presence of guideline-recommended ST-segment elevation (STE) criteria on electrocardiogram (ECG) readings [21].

During initial clinical evaluation, investigators collected data from all patients regarding characteristics, onset of chest pain and additional symptoms, and presence of risk factors for ACS (HTA, DM, HLP, heredity, smoking). Smoking history was obtained from all participants as follows: non-smokers and smokers (including current and ex-smokers). All patients experienced acute chest pain with maximal one hour duration prior to admission. Diagnosis of STEMI or NSTEMI was established in regard to clinical presentation, standard 12-lead electrocardiography (ECG) interpretation, and cardiac enzyme measurements (high-sensitivity troponin I (hsTnI), proBNP, creatine kinase-MB (CK-MB)), which were all interpreted according to the previously established criteria of the European Society of Cardiology [22,23]. For all ACS patients, standard transthoracic echocardiography (TTE) was performed on the first day of hospitalization, after admission. TTE included: measurements of left and right chamber dimensions in M-mode, estimation of left ventricular (LV), ejection fraction (EF) assessed by the biplane Simpson method, estimation of left ventricle segment kinetics in 2D mode, and wall motion score index (WMSI). Invasive coronary angiography was performed in all patients in accordance with ESC guidelines [21]. After initial evaluation of the invasive coronary angiography, the decision concerning further treatment was made (percutaneous or surgical coronary revascularization or extension of optimal pharmacological therapy).

The main inclusion criteria were voluntary informed and written participation, ACS diagnosis, and age above 18 years old.

The main exclusion criteria for study participation were as follows: (1) patients younger than 18 or older than 80 years; (2) pregnant and breastfeeding women; (3) patients with significant cognitive deficits; (4) patients with periprocedural infarction, stent thrombosis, or secondary infarction; and (5) patients using medications with antioxidant properties.

### 2.3. Determination of Cardiac Enzymes at Admission and 12 h after Admission

Blood samples were collected from all patients from the antecubital vein for measurement of cardio-specific enzymes. Levels of high-sensitivity troponin I (hsTnI), N-terminal (NT)-pro hormone BNP (NT-pBNP), and creatine phosphokinase-MB (CK-MB) were measured at admission (not longer than one hour from chest pain onset) (T0) and 12 h after admission (T12). Cardiac enzymes were measured by chemiluminescent immunoassay in a modification of the standard enzyme immunoassay (ECLIA) and presented in nanograms per milliliter (ng/mL).

### 2.4. Assessment of Biomarkers of Oxidative Stress at Admission, 6 h, and 12 h after Admission

Blood samples for evaluating oxidative stress biomarkers were taken from the antecubital vein at three time points: at admission (not longer than one hour from chest pain onset) before any therapy administration (T0), 6 h (T6), and 12 h after admission (T12) for all STEMI and NSTEMI patients.

In the collected venous blood samples, the following OS markers were spectrophotometrically measured: (1) index of lipid peroxidation (measured as TBARS—thiobarbituric acid reactive substances), (2) nitrites (NO_2_−) as marker of NO outflow, (3) hydrogen peroxide (H_2_O_2_), and (4) superoxide anion radical (O_2_−). The antioxidative defense system was estimated by determination of antioxidative enzymes in erythrocytes: superoxide dismutase (SOD), catalase (CAT), and reduced glutathione (GSH).

### 2.5. Determination of Superoxide Anion Radical (O_2_-)

Superoxide anion radical (O_2_−) levels were estimated using a reaction of nitro blue tetrazolium (NBT) in TRIS buffer along with the plasma samples, measured at 530 nm. A distilled water solution was used for blank probes [23].

### 2.6. Determination of Hydrogen Peroxide (H_2_O_2_)

Hydrogen peroxide (H_2_O_2_) was measured via oxidation of phenol red by hydrogen peroxide in the presence of horseradish peroxidase (HRPO). We combined 200 μL of plasma with 800 μL of phenol red solution (PRS) and 10 μL of (1:20) HRPO. The level of H_2_O_2_ was measured at 610 nm. A distilled water solution was used for the blank probe. The level of H_2_O_2_ was measured at 610 nm [24].

### 2.7. Determination of Nitrites (NO_2_−)

After rapid decomposition, nitric oxide (NO) forms a stable nitrite/nitrate metabolite product. Levels of nitrite (NO_2_−) and nitrate were detected using a method based on the Griess reaction. Using a Griess reagent, NO_2_− was determined as an index of NO production. A combination of 0.1 mL 3 N perchloride acid, 0.4 mL 20 mmol/L ethylenediaminetetraacetic acid (EDTA), and 0.2 mL of plasma were settled on ice for 15 min and then centrifuged at 6000 rpm for 15 min. After the supernatant was poured off, 220 μL K_2_CO_3_ was added. NO_2_− was measured at 550 nm. A distilled water solution was used for the blank probe [25].

### 2.8. Determination of Index of Lipid Peroxidation Measured as TBARS

The lipid peroxidation level was estimated by measuring TBARS, using 1% thiobarbituric acid (TBA) in 0.05 sodium hydroxide (NaOH), which was incubated with 0.8 mL of plasma at 100 °C for 15 min and then measured at 530 nm. A distilled water solution with 1% TBA in 0.05 NaOH was used for the blank probe [26].

### 2.9. Assessment of Activity of Antioxidant Enzymes at Admission, 6 h, and 12 h after Admission

Isolated erythrocytes were washed 3 times with 3 volumes of 0.9 mmol/L sodium chloride (NaCl). Determination of CAT activity was performed by using hemolysate [27], which contained 50 g of Hgb/L. After being diluted in distilled water (1:7 *v*/*v*), lysates were combined with chloroform ethanol (0.6:1 *v*/*v*) in order to remove hemoglobin [28]; 50 μL CAT buffer, 100 μL sample, and 1 mL 10 mM H2O2 were added to the samples and measured at 360 nm. A distilled water solution was used for the blank probe. Determination of superoxide dismutase (SOD) activity was performed using the epinephrine method [29]. A total of 100 μL of epinephrine was combined with a mixture of 100 μL lysates and 1 mL of bicarbonate buffer. The measurement of SOD activity was made at 470 nm. The reduced glutathione (GSH) was estimated by GSH oxidation with 5.5-dithiobis-6.2-nitrobenzoic acid. Extract of GSH was obtained by combining 0.1 mL 0.1% EDTA, 400 μL plasma, and 750 μL precipitation solution (containing 1.67 g metaphosphoric acid, 0.2 g EDTA, 30 g NaCl, and filled with distilled water until 100 mL; the solution is stable for 3 weeks at +4 °C). After mixing in the vortex machine (MX-E 3000 rpm laboratory Fixed Speed Lab Vortex Mixer, Drawell, China) and extraction on cold ice for 15 min, it was centrifuged at 4000 rpm for 10 min. A distilled water solution was used for the blank probe. The measurement of reduced GSH was made at 420 nm [30].

### 2.10. Statistical Analyses

Continuous variables were presented as mean ± standard deviation, while categorical variables were described as frequencies and percentages. Differences between the patient groups for categorical variables were examined using the χ^2^, Fisher exact, or z test. Differences in the continuous variables between groups were assessed using the Wilcoxon rank sum test where appropriate. A univariate logistic model was used to examine the relationship between all variables. After each variable was tested independently in a univariate regression model, those that achieved a value of *p* < 0.25 (and were clinically meaningful) were selected for testing in a multivariable logistic regression. We used 95% CIs to illustrate the association between potential variables and in-hospital mortality. A two-sided *p* value < 0.05 was considered statistically significant. SPSS V.26.0 was used to perform all statistical analyses (SPSS Inc., Chicago, IL, USA).

## 3. Results

### 3.1. Basic Characteristics of Study Population

Between January 2016 and January 2017, 99 participants were included in the study. From this cohort, 40 patients presented with STEMI and 39 patients with non-STEMI. Table 1 summarizes the baseline characteristics of all patients. The study population was predominately female (52%), and there were no statistically significant differences in distribution of male or female (Table 1). The mean age was 56.17 ± 1.22 in the STEMI group and 69.17 ± 3.65 in the non-STEMI group; all patients had a previous history of HTA, CVD, DM, and/or DYS. Most of the participants were also smokers. According to the Killip classification, most of the patients were at the second and third level (Table 1).

### 3.2. Cardiac Enzymes at Admission and 12 h after Admission

Table 2 shows the means of concentration of main cardiac enzymes at two points. In comparing the values at admission and 12 h after, all cardiac enzymes were statistically significantly different (Table 2).

### 3.3. Levels of Pro-Oxidative Plasma Biomarkers in Study Groups

Figure 1, Figure 2, Figure 3 and Figure 4 show the mean values of plasma concentration of oxidative markers measured at admission, 6 h, and 12 h after admission. Figure 1 shows that the level of superoxide anion radicals was statistically significantly increased in the non-STEMI group in comparison with the STEMI and CTRL groups 6 h after admission, but decreased 12 h after admission (Figure 1). Levels of hydrogen peroxide are presented in Figure 2. For all time points, this marker was statistically significantly increased in the non-STEMI group in comparison with the other groups (Figure 2).

Similarly, bioavailability of nitric oxide was significantly decreased in STEMI group, and increased in the non-STEMI group (Figure 3). In addition, the index of lipid peroxidation was significantly increased in the non-STEMI group (Figure 4).

### 3.4. Antioxidant Capacity of Patients with STEMI/Non-STEMI Presentation

Antioxidant capacity was evaluated by the activity of CAT, GSH, and SOD enzymes in hemolysate of patients at three time points. Figure 5, Figure 6 and Figure 7 represent the dynamics of the mentioned markers. Catalase activity was significantly increased in the STEMI group 6 h and 12 h after admission, while the content of reduced glutathione was not altered in the STEMI and non-STEMI groups, but was increased in comparison with the CTRL group (Figure 6). Lastly, SOD activity was significantly increased in the non-STEMI group (Figure 7).

The ANOVA table presents the differences in all biomarkers of oxidative stress in relation to group and time of sampling. We observed that almost all markers, such as hydrogen peroxide, nitric oxide, TBARS, and all antioxidant enzymes were different in relation to group (STEMI vs. non-STEMI). On the other hand, taking together all values, time of sampling was significant only for antioxidant enzymes (Table 3).

### 3.5. Correlation Analysis of the Measured Biomarkers of Redox Status and Cardiac Enzymes in Patients with STEMI and Non-STEMI Manifestation of Disease

Table 4 presents the correlation analysis of biomarkers of oxidative stress and cardiac enzymes at different time points. We observed a significant positive correlation of superoxide anion radicals and levels of troponin I at admission and an inverse correlation between reduced glutathione and levels of NT-pBNP measured 6 h after admission (Table 4).

Figure 8 and Figure 9 present the dynamic of superoxide anion radicals and reduced glutathione in relation to correlated cardiac enzymes as predictors of an STEMI or non-STEMI course. Both of the presented markers, superoxide anion radicals and reduced glutathione, could be an early predictor of reperfusion injury and indicators of difference in ST elevation (STEMI or NSTEMI).

## 4. Discussion

This prospective observational case-control study included 79 patients with acute coronary syndrome (ACS), which refers to any constellation of clinical symptoms that are compatible with acute myocardial ischemia. ACS is divided into ST-elevated myocardial infarction (STEMI), non-ST-elevated myocardial infarction (NSTEMI), and unstable angina (UA) [31]. Since this illness causes very serious conditions that could have fatal outcomes, it is very important to find the predictive factors that contribute to the pathogenesis of these ischemic events. A non-ST elevation myocardial infarction is a type of heart attack that usually happens when the heart’s need for oxygen can not be met, and this life-threating condition is very often undetected because of its difficult-to-identify electrical pattern. Therefore, the aim of our study was to determine the role of OS during early evaluation of acute STEMI and NSTEMI patients in order to define the role of redox balance in profiling the development of MI. In addition, we wanted to obtain information about the possible predictors for developing STEMI or non-STEMI.

On the other hand, previous literature data has suggested that oxidative stress is associated with ST elevation and some clinical measures in patients with myocardial infarction [32]. As one of the most prevalent forms of ischemic heart diseases, acute myocardial infarction is based on reperfusion of the myocardium. Actually, during the reperfusion period caused by blood flow after MI, huge quantities of reactive oxygen species (ROS) are produced [33]. During these events, reperfusion injury could be very strong, caused by highly reactive oxygen species that paradoxically lead to a reduction in the beneficial effects of post-ischemic reperfusion. In this moment, ST resolution is often an indicator of the epicardial coronary flow [34].

In this study, we investigated the possible role and dynamics of oxidative stress markers (superoxide anion radicals, hydrogen peroxide, index of lipid peroxidation, nitric oxide) and antioxidative enzymes among STEMI and non-STEMI patients.

At first, we evaluated the dynamic of the superoxide anion and hydrogen peroxide markers’ concentrations. As is known, the production of superoxide anion radicals is essential for an aerobic organism. These pro-oxidative markers act as signaling molecules in various processes such as aging and apoptosis [35]. In addition, in many chronic diseases such as diabetes and ischemic vascular disease, superoxide anion radicals are important biomolecules that can alter physiological function [35]. In our study, we confirmed the elevation of superoxide anion radicals in the non-STEMI group in comparison with the STEMI and CTRL groups 6 h after admission, but they decreased 12 h after admission (Figure 1). Furthermore, hydrogen peroxide is close in function to superoxide anion radicals and causes their production. Levels of hydrogen peroxide are presented in Figure 2. In non-STEMI group, this marker was statistically significantly increased in comparison with other groups at all time points (Figure 2).

A previous study conducted by Nesim et al. investigated the influence of oxidants and pro-oxidants in myocardial infarction [36]. They prospectively followed 50 STEMI and 55 NSTEMI patients to measure pro-oxidants and antioxidants. Their results showed that patients with ACS had a significantly greater index of lipid peroxidation and lower levels of SOD and CAT. The results of their study confirmed that patients with STEMI and NSTEMI had greater oxidative stress [36].

In addition to superoxide anion radicals and hydrogen peroxide levels, we measured levels of nitric oxide and the index of lipid peroxidation (measured as TBARS) from plasma samples (Figure 3 and Figure 4). The bioavailability of nitric oxide was significantly decreased in the STEMI group and increased in the non-STEMI group (Figure 3). In addition, the index of lipid peroxidation was significantly increased in the non-STEMI group (Figure 4) in comparison with the CTRL groups. These results were expected, and in all ACS patients we confirmed the high production of systemic oxidative stress at admission and during hospital treatment.

Cardiac ischemia causes oxidative stress, leading to the generation of reactive species which can modify cellular structures such as proteins or nucleic acids. For us, the index of lipid peroxidation levels was extremely important. Thiobarbituric-acid reactive substances (TBARS), as a lipid peroxidation product, is one of the first well-known oxidative stress biomarkers. Similar to malondialdehyde acid, it is generated when oxygen-derived free radicals surround and break down polyunsaturated fatty acids. That means that this marker is a direct risk factor for coronary artery disease severity and plaque sensitivity [37]. In addition, elevated lipid peroxidation index levels were confirmed in ACS patients in comparison with healthy participants (Figure 1, Figure 2, Figure 3 and Figure 4).

In the second part, we observed the dynamic of antioxidant enzymes, such as superoxide dismutase (SOD), catalase (CAT) and reduced glutathione (GSH) in patients with ACS. Antioxidant capacity was evaluated by the activity of CAT, GSH, and SOD enzymes in the hemolysate of patients at three time points. Figure 5, Figure 6 and Figure 7 represent the dynamic of the mentioned markers. Catalase activity was significantly increased in the STEMI group 6 h and 12 h after admission, while the content of reduced glutathione was not altered in the STEMI and non-STEMI groups but was increased in comparison with the CTRL group (Figure 6). Lastly, SOD activity was significantly increased in the non-STEMI group (Figure 7). Superoxide dismutase and glutathione peroxidase are the first well-known line of antioxidant enzymes. Previous studies showed that oxidant–antioxidant imbalance is responsible for the pathogenesis of myocardial ischemia and reperfusion injury [38,39]. In our study, we obtained different results for the activity of some types of enzymes, and over time and hospital treatment we observed higher levels of SOD and CAT. This could be a consequence of reperfusion therapy, and it was indicated that the increased activity of total antioxidant capacity reduces the damage produced by the enhancement of lipid peroxidation, which may act by counteracting its harmful products. We know that SOD converts superoxide anion radicals to hydrogen peroxide, which is then reduced to water and oxygen by catalase activity. Since the patients with STEMI and NSTEMI underwent therapy, it was expected that at 6 h and 12 h after admission they would have elevated levels of the enzymes. Usually, in clinical situations with ACS, oxidative stress increases while antioxidant status decreases. The disruption in blood flow and endothelial dysfunction may lead to the formation of plaques in the coronary arteries, and all these processes are caused by oxidative stress. In this regard, reactive oxygen species are initiators, but SOD and GPx are the most important enzymes in supporting and protecting cells against damage. A previous study indicated that PCI could induce more damage, and that could be a reason for decreased antioxidant capacity in our patients [40]. This knowledge is very important to us, and it means that patients who undergo PCI must be supplemented with more antioxidants or products that could improve antioxidant capacity in the early period after myocardial infarction. Improving antioxidant capacity could also reduce the incidence of some complications such as arrhythmia and mortality [41].

In order to completely confirm the role of oxidative stress markers in patients with myocardial infarction and different clinical presentations, we performed correlation analysis of all biomarkers and cardiac enzymes in the study groups. Table 4 presents the correlation analysis of biomarkers of oxidative stress and cardiac enzymes over time. We observed a significant positive correlation of superoxide anion radicals and levels of troponin I at admission and an inverse correlation between reduced glutathione and levels of NT-pBNP measured 6 h after admission (Table 4). Figure 8 and Figure 9 present the dynamic of superoxide anion radicals and reduced glutathione in relation to correlated cardiac enzymes as predictors of a STEMI or non-STEMI course. Both markers, superoxide anion radicals and reduced glutathione, could be an early predictor of reperfusion injury and an indicator of difference in ST elevation (STEMI or NSTEMI).

A previous study also evaluated potential predictors of acute coronary disease and potential complications. Lubrano et al. compared the levels of recently proposed OS-related parameters in acute coronary syndrome (ACS) and stable coronary artery disease (CAD) to evaluate their effectiveness as additive risk or illness indicators of stable and acute ischemic events and their response over time during the course of AMI. These authors concluded that different OS-related biomarkers were differentially associated with CV risk factors and CAD or ACS presence [42].

Matin et al. investigated the levels of oxidative stress markers and their association with clinical outcomes in patients with ST-segment elevation myocardial infarction (STEMI) undergoing primary percutaneous coronary intervention [42]. This study indicated that antioxidant enzyme levels in patients with STEMI are significantly associated with coronary artery stenosis and the level of response to treatment. However, such relationships were not observed regarding TAC and MDA levels [43].

Aksoy et al. evaluated the association between left ventricular ejection fraction recovery and total oxidant status, total antioxidant capacity, and high-sensitivity C-reactive protein levels [43]. They confirmed that oxidative stress and inflammation parameters were detrimental to the recovery of left ventricular ejection fraction in patients with ST-elevation myocardial infarction [43].

Our results are in accordance with the results of previous studies that highlight higher oxidative stress status in both STEMI and NSTEMI patients compared with healthy participants. In addition, our study confirmed the link between oxidative stress and presence of CV risk factors, as have a number of previous studies. However, very importantly, for the first time our study emphasized the exact markers that directly contribute to the pathophysiology of myocardial infarction. The superoxide anion radicals and reduced glutathione observed with cardiac enzymes could definitely allow early prediction of further onset of ST elevation.

Limitations of this research lie in the absence of measuring of inflammatory factors in patients with STEMI and NSTEMI. We should be able to provide more detailed information about the pathogenesis by using the dynamic of inflammation and linking it with oxidative stress, but from this point of view, we can also assume that inflammation acts very similarly to oxidative stress, with the same dynamic. Nevertheless, our approach for observing the oxidative stress dynamic in patients with ACS is sufficient for this type of prediction.

## 5. Conclusions

Differences in oxidative stress-related biomarkers (between groups, according to response over time at admission, and during hospital treatment) confirmed oxidative stress involvement in the transition from healthy status to STEMI and NSTEMI, although evidencing the heterogeneous nature of redox processes. We confirmed that superoxide anion radicals and reduced glutathione observed together with hs-troponin I at admission and NT-pBNP during hospital treatment could be predictors of ST evolution. In future, a multi-marker panel including different biomarkers and pathways of oxidative stress could be evaluated as an additive tool to be used in CV prediction, diagnosis, patient stratification, and treatment.

## Figures and Tables

**Figure 1 jpm-13-01050-f001:**
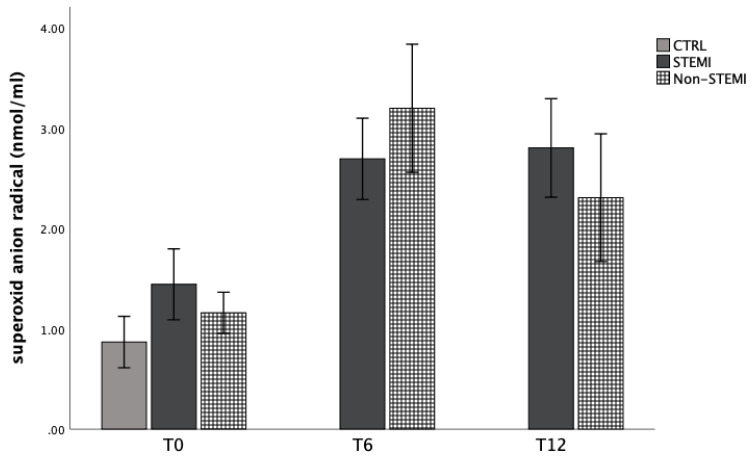
Plasma concentration of superoxide anion radicals among STEMI, non-STEMI, and healthy participants measured at admission (T0), 6 h after admission (T6), and 12 h after admission (T12). Results are presented as mean ± standard deviation (X ± SD).

**Figure 2 jpm-13-01050-f002:**
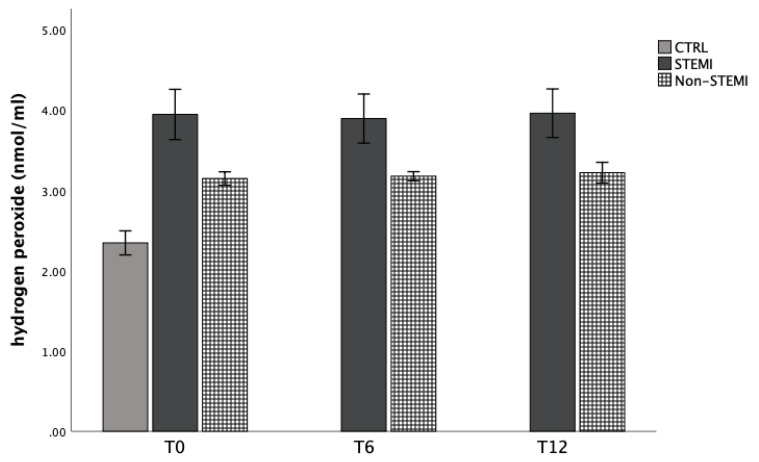
Plasma concentration of hydrogen peroxide among STEMI, non-STEMI, and healthy participants measured at admission (T0), 6 h after admission (T6), and 12 h after admission (T12). Results are presented as mean ± standard deviation (X±SD).

**Figure 3 jpm-13-01050-f003:**
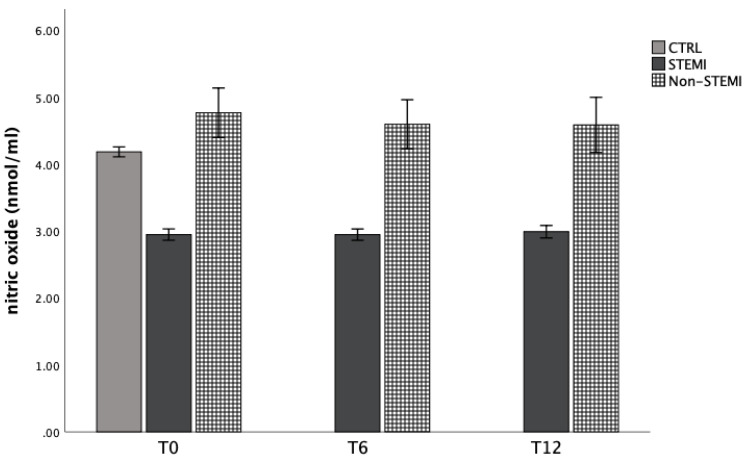
Plasma concentration of nitric oxide measured as nitrites among STEMI, non-STEMI, and healthy participants measured at admission (T0), 6 h after admission (T6), and 12 h after admission (T12). Results are presented as mean ± standard deviation (X ± SD).

**Figure 4 jpm-13-01050-f004:**
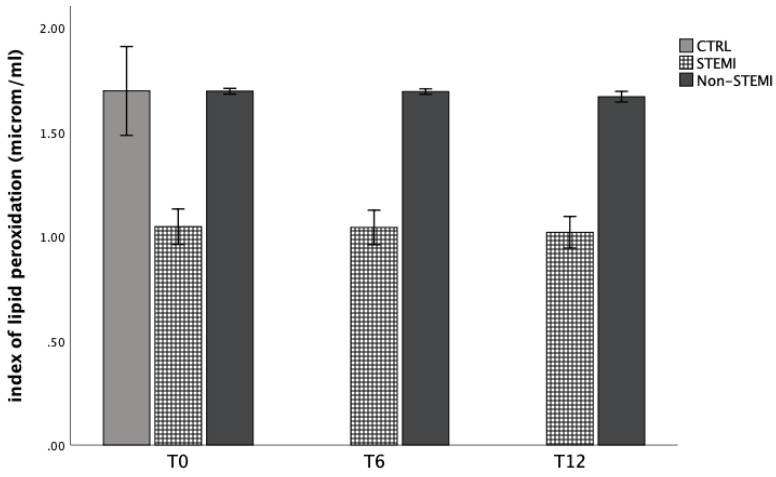
Plasma concentration of index of lipid peroxidation measured as TBARS among STEMI, non-STEMI, and healthy participants measured at admission (T0), 6 h after admission (T6), and 12 h after admission (T12). Results are presented as mean ± standard deviation (X ± SD).

**Figure 5 jpm-13-01050-f005:**
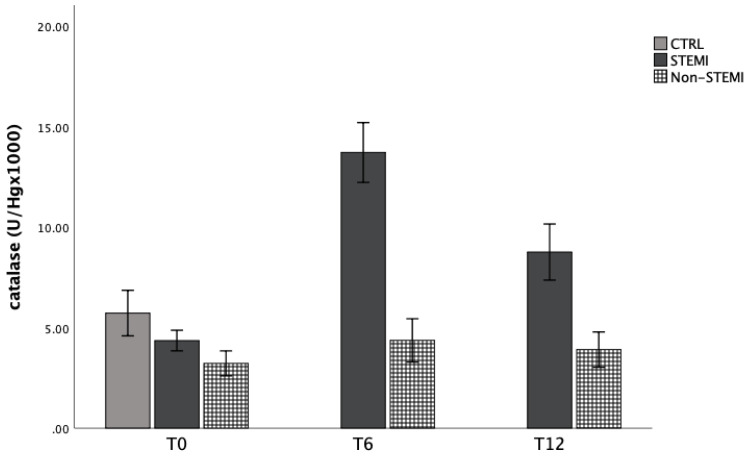
Catalase activity in hemolysate of STEMI, non-STEMI, and healthy participants measured at admission (T0), 6 h after admission (T6), and 12 h after admission (T12). Results are presented as mean ± standard deviation (X ± SD).

**Figure 6 jpm-13-01050-f006:**
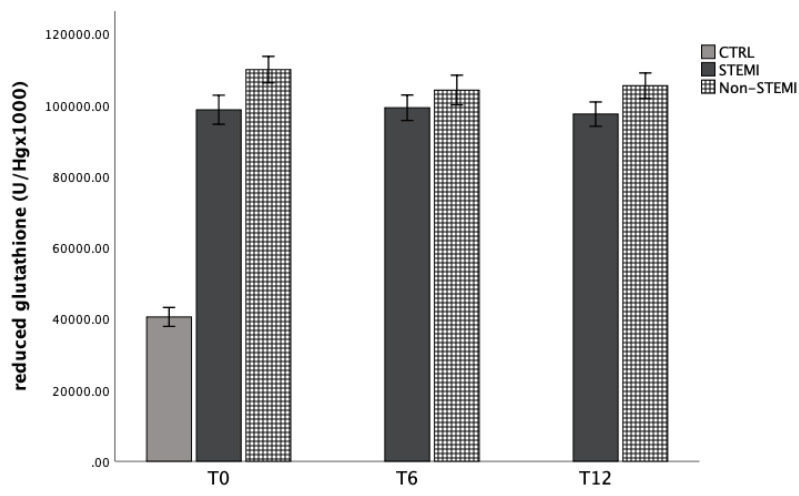
Reduced glutathione content in hemolysate of STEMI, non-STEMI, and healthy participants measured at admission (T0), 6 h after admission (T6), and 12 h after admission (T12). Results are presented as mean ± standard deviation (X ± SD).

**Figure 7 jpm-13-01050-f007:**
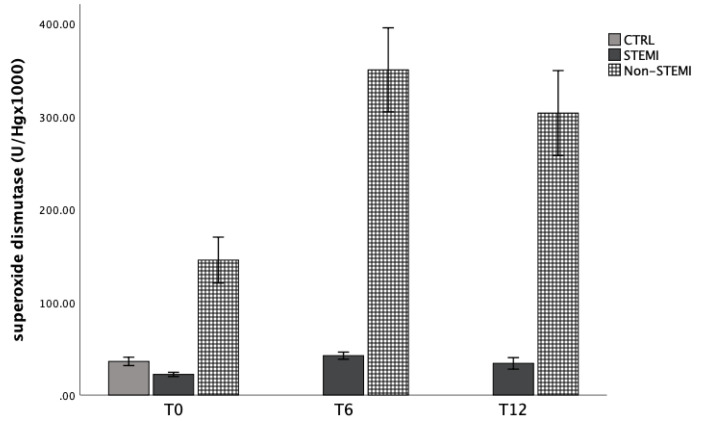
Superoxide dismutase activity in hemolysate of STEMI, non-STEMI, and healthy participants measured at admission (T0), 6 h after admission (T6), and 12 h after admission (T12). Results are presented as mean ± standard deviation (X ± SD).

**Figure 8 jpm-13-01050-f008:**
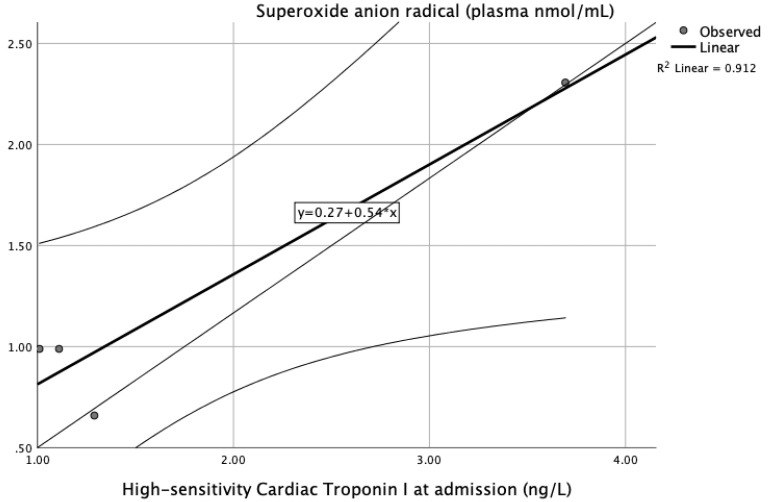
ROC curve of significant variables O_2_− and hsTPI of STEMI, non-STEMI, and healthy participants.

**Figure 9 jpm-13-01050-f009:**
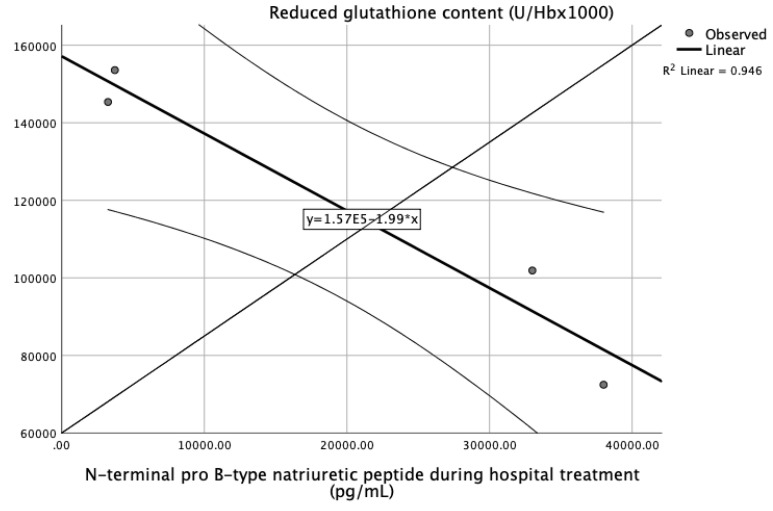
ROC curve of significant variables GSH and NT-pBNP of STEMI, non-STEMI, and healthy participants.

**Table 1 jpm-13-01050-t001:** Baseline characteristics of the study population.

Variables	CTRL Group (*n* = 20)	STEMI Group (*n* = 40)	Non-STEMI Group (*n* = 39)	Statistical Significance
Age (years)	65.11 ± 2.34	56.17 ± 1.22	69.17 ± 3.65	*p* > 0.674
Gender (F/M%)	10 (50%); 10 (50%)	23 (57.5%); 17 (42.5%)	19 (48.7%); 20 (51.3%)	*p* > 0.761
Time to hospital admission (min)	/	59.09 ± 0.11	61.02 ± 0.23	*p* > 0.709
Hypertension (yes%)	0 (0%)	40 (100%)	39 (100%)	*p* > 0.623
Dyslipidemia (yes%)	0 (0%)	40 (100%)	39 (100%)	*p* > 0.611
Diabetes mellitus (yes%)	0 (0%)	40 (100%)	39 (100%)	*p* > 0.782
Previous CVD (yes%)	0 (0%)	40 (100%)	39 (100%)	*p* > 0.881
Smoking (yes%)	17 (85%)	22 (55%)	26 (66.7%)	*p* > 0.578
Killip classification	/			*p* > 0.604
Class I (*n*,%)		0 (0%)	1 (2.6%)	
Class II (*n*,%)		10 (25%)	13 (33.3%)	
Class III (*n*,%)		15 (37.5%)	12 (30.8%)	
Class IV (*n*,%)		15 (37.5%)	23 (58.9%)	

**Table 2 jpm-13-01050-t002:** The mean concentrations of cardiac enzymes at admission and during hospital treatment (12 h after admission) in the STEMI and non-STEMI groups. Results are presented as mean ± standard deviation (X ± SD). Statistical significance was confirmed using the Wilcoxon test for repeated measured samples.

	hsTnI (ng/L) at T0 (X ± SD)	hsTnI (ng/L) at T12 (X ± SD)	CKMB (IU/L) at T0 (X ± SD)	CKMB (IU/L) at T12 (X ± SD)	NT-pBNP (pg/L) at T0 (X ± SD)	NT-pBNP (pg/L) at T12 (X ± SD)
STEMI	3.69 ± 0.89	26.03 ± 2.13	40.17 ± 7.16	118.27 ± 8.96	936.91 ± 64.23	3724.67 ± 256.32
Non-STEMI	1.01 ± 0.35	5.92 ± 0.78	28.03 ± 5.23	53.45 ± 6.12	861.16 ± 104.23	3244.64 ± 151.23
Statistical significance (*p* value)	*p* = 0.032 *	*p* = 0.002 *	*p* = 0.022 *	*p* = 0.027 *	*p* = 0.062	*p* = 0.041 *

* value less than 0.05.

**Table 3 jpm-13-01050-t003:** ANOVA table and statistical comparisons in relation to group (CTRL, STEMI, non-STEMI) or to time of measurement (T0, T6, or T12).

	Comparisons		*p* Value	Comparisons		*p* Value
O_2_- (nmol/mL)	T0	T6	0.275	STEMI	Non-STEMI	0.960
		T12	0.830		CTRL	0.100
	T6	T0	0.275	Non-STEMI	STEMI	0.960
		T12	0.750		CTRL	0.134
	T12	T0	0.830	CTRL	STEMI	0.100
		T6	0.750		Non-STEMI	0.134
H_2_O_2_ (nmol/mL)	T0	T6	0.955	STEMI	Non-STEMI	0.611
		T12	0.716		CTRL	0.009 *
	T6	T0	0.955	Non-STEMI	STEMI	0.611
		T12	0.883		CTRL	0.033 *
	T12	T0	0.716	CTRL	STEMI	0.009 *
		T6	0.883		Non-STEMI	0.033 *
NO^−^ (nmol/mL)	T0	T6	0.923	STEMI	Non-STEMI	0.000 **
		T12	0.508		CTRL	0.980
	T6	T0	0.923	Non-STEMI	STEMI	0.000 **
		T12	0.300		CTRL	0.015 *
	T12	T0	0.508	CTRL	STEMI	0.980
		T6	0.300		Non-STEMI	0.015 *
TBARS (μmol/mL)	T0	T6	0.817	STEMI	Non-STEMI	0.000 **
		T12	0.619		CTRL	0.000 **
	T6	T0	0.817	Non-STEMI	STEMI	0.000 **
		T12	0.946		CTRL	0.950
	T12	T0	0.619	CTRL	STEMI	0.000 **
		T6	0.946		Non-STEMI	0.950
SOD (U/Hg × 10^3^)	T0	T6	0.023 *	STEMI	Non-STEMI	0.000 **
		T12	0.072		CTRL	0.949
	T6	T0	0.023 *	Non-STEMI	STEMI	0.000 **
		T12	0.000 *		CTRL	0.000 **
	T12	T0	0.072	CTRL	STEMI	0.949
		T6	0.000 *		Non-STEMI	0.000 **
GSH (U/Hg × 10^3^)	T0	T6	0.620	STEMI	Non-STEMI	0.071
		T12	0.000 *		CTRL	0.006
	T6	T0	0.620	Non-STEMI	STEMI	0.071
		T12	0.001 **		CTRL	0.101
	T12	T0	0.000 **	CTRL	STEMI	0.006 *
		T6	0.001 **		Non-STEMI	0.101
CAT (U/Hg × 10^3^)	T0	T6	0.000 **	STEMI	Non-STEMI	0.000 **
		T12	0.045 *		CTRL	0.024 *
	T6	T0	0.000 **	Non-STEMI	STEMI	0.000 **
		T12	0.049 *		CTRL	0.751
	T12	T0	0.045 *	CTRL	STEMI	0.024 *
		T6	0.049 *		Non-STEMI	0.751

* *p*-statistical significance with statistical threshold ≤0.05; ** *p*-statistical significance with statistical threshold ≤0.001.

**Table 4 jpm-13-01050-t004:** Correlation matrix. Significant correlation was confirmed if *p* values were equal or below 0.05; r coefficient represents size and direction of correlation.

Markers		O_2_− (nmol/mL)	H_2_O_2_ (nmol/mL)	NO^−^ (nmol/mL)	TBARS (μmol/mL)	CAT (U/Hg × 10^3^)	GSH (U/Hg × 10^3^)	SOD (U/Hg × 10^3^)
O_2_− (nmol/mL)	r	1	−0.076	0.148 *	0.069	0.120	0.078	0.023
	*p*		0.309	0.047	0.358	0.109	0.300	0.764
H_2_O_2_ (nmol/mL)	r	−0.076	1	−0.268 **	−0.407 **	−0.069	−0.029	−0.156 *
	*p*	0.309		0.000	0.000	0.319	0.670	0.024
NO^−^ (nmol/mL)	r	0.148 *	−0.268 **	1	0.323 **	−0.005	−0.083	0.345 **
	*p*	0.047	0.000		0.000	0.941	0.227	0.000
TBARS (μmol/mL)	r	0.069	−0.407 **	0.323 **	1	−0.112	0.204 **	0.420 **
	*p*	0.358	0.000	0.000		0.105	0.003	0.000
CAT (U/Hg × 10^3^)	r	0.120	−0.069	−0.005	−0.112	1	0.118	−0.230 **
	*p*	0.109	0.319	0.941	0.105		0.086	0.001
GSH (U/Hg × 10^3^)	r	0.078	−0.029	−0.083	0.204 **	0.118	1	0.034
	*p*	0.300	0.670	0.227	0.003	0.086		0.626
SOD (U/Hg × 10^3^)	r	0.023	−0.156 *	0.345 **	0.420 **	−0.230 **	0.034	1
	*p*	0.764	0.024	0.000	0.000	0.001	0.626	
hsTnI (ng/L) at admission	r	0.955 *	−0.604	−0.789	−0.291	0.900	0.583	−0.483
	*p*	0.045	0.396	0.211	0.709	0.100	0.417	0.517
CKMB (IU/L) at admission	r	0.642	0.318	−0.144	0.633	0.859	−0.174	−0.180
	*p*	0.358	0.682	0.856	0.367	0.141	0.826	0.820
NT-pBNP (pg/L) at admission	r	0.677	0.229	−0.250	0.563	0.891	−0.148	−0.151
	*p*	0.323	0.771	0.750	0.437	0.109	0.852	0.849
hsTnI (ng/L) during hospital treatment	r	0.626	0.353	−0.099	0.660	0.843	−0.183	−0.193
	*p*	0.374	0.647	0.901	0.340	0.157	0.817	0.807
CKMB (IU/L) during hospital treatment	r	0.650	0.340	−0.093	0.645	0.854	−0.142	−0.240
	*p*	0.350	0.660	0.907	0.355	0.146	0.858	0.760
NT-pBNP (pg/L) during hospital treatment	r	−0.619	0.572	0.202	0.558	−0.305	−0.973 *	0.927
	*p*	0.381	0.428	0.798	0.442	0.695	0.027	0.073

* *p*-statistical significance with statistical threshold ≤0.05; ** *p*-statistical significance with statistical threshold ≤0.001.

## Data Availability

All published research data are available on request to the corresponding author.

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
