# Peer review of "Prognostic Value of Redox Status Biomarkers in Patients Presenting with STEMI or Non-STEMI: A Prospective Case-Control Clinical Study"

_jpm, 2023, doi:10.3390/jpm13071050_

Round 1

Reviewer 1 Report

1- Please spell the abbreviations first in the abstract and the manuscript

2- This data is from 2016-2017, why wasn't it published earlier? I don't see any long-term follow-up, so what was the cause of the delay?

3- Several studies was published before on the role of oxidative stress in MI, what is the novel thing about this study.

4- Do you have any evidence from your data that the administration of anti oxidants would have a significant effect on the outcomes?

5- Discussion is very long. Consider making it more comprehensive with 1-2 studies supporting and/or conflicting and add an explanation why do you think they are similar/not similar to your results. Simply stating what was the old studies saying does not add any scientific value to the discussion

Quality of English is appropriate, could use some moderate editing.

Author Response

R1

  • Please spell the abbreviations first in the abstract and the manuscript
  • A: Corrected Thank you.
  • This data is from 2016-2017, why wasn't it published earlier? I don't see any long-term follow-up, so what was the cause of the delay?
  • Thank you for this question. Honestly, our first author Dr Zorica had a serious health problems and because of that all data and analyzing waiting so long. We hope that there is still enough space to publish our results. Thank you for understanding.

3- Several studies was published before on the role of oxidative stress in MI, what is the novel thing about this study.

A: The novel thing is that we calculated the exactly biomarkers who could be measured routinely and be a significant predictor of negative outcome. We confirmed that superoxide anion radical and reduced glutathione observed together with hs-Troponin I at admission and NT-pBNP during hospital treatment could be a predictors of ST evolution-that is the main novelty of this study.

  • Do you have any evidence from your data that the administration of anti oxidants would have a significant effect on the outcomes?

A: In our study we very carefully chosen the patients who were without any additional antioxidant treatment in the last three months. All patients with prescribed for example omega-3 fatty acids, high doses of vitamin C or D, plant extracts in high concentrations or on special dietary treatments were excluded prioir to study.

  • Discussion is very long. Consider making it more comprehensive with 1-2 studies supporting and/or conflicting and add an explanation why do you think they are similar/not similar to your results. Simply stating what was the old studies saying does not add any scientific value to the discussion

A: Thank you. Corrected.

Comments on the Quality of English Language

Quality of English is appropriate, could use some moderate editing.

A: Thank you. Corrected.

R2

Dear Authors,

I would like to congratulate you with the results of your study. A: Thank you.

I would like to thank you for having chance to review the manuscript you've submitted. A: Thank you.

I have some concerns and suggestions that I would like to share with you:

  1. In Abstract you did not mention in results section any exact numbers that is required. The abstract is too descriptive and vaque.

A: Thank you. Corrected. Please abstract.

  1. In Table 1 - please provide exact p values
  2. A: Thank you. Corrected.
  3. Figures 1-4, the p values between groups and certain results in presented T0,T6, T12 are required, I would suggest to add it into the text and Figures. Otherwise, the results and potencial statistical differences between groups are not clear
  4. In the form of the Table 3, you can see the all statiscal values which are related to mentioned Figures. Please see Table 3. ANOVA table and statistical comparisons in relation to group (CTRL, STEMI, Non-STEMI) or to point of measurement (T0, T6 or T12).
  5. Figures 5-7 - the same suggestions as above
  6. In the form of the Table 3, you can see the all statiscal values which are related to mentioned Figures. Please see Table 3. ANOVA table and statistical comparisons in relation to group (CTRL, STEMI, Non-STEMI) or to point of measurement (T0, T6 or T12).

5.In Table 4, as I found only strong correlation between TnI on admission and oxygen radicals (02-) followed by NT-proBNP and GSH, this results should be underlined in your results section. According to Table 2 only TNI on admission was signifcantly different between STEMI and nonSTEMI groups. Please, try to follow that results in discussion and conclusion. I suggest to reconsider the title, discussion and conclusion basing on the presented results.

A: We are in general focused on that results, please Abstract, Results and Dicussion.

Kinds

Reviewer

Reviewer 2 Report

Dear Authors,

I would like to congratulate you with the results of your study.

I would like to thank you for having chance to review the manuscript you've submitted.

I have some concerns and suggestions that I would like to share with you:

1. In Abstract you did not mention in results section any exact numbers that is required. The abstract is too descriptive and vaque.

2. In Table 1 - please provide exact p values

3. Figures 1-4, the p values between groups and certain results in presented T0,T6, T12 are required, I would suggest to add it into the text and Figures. Otherwise, the results and potencial statistical differences between groups are not clear

4. Figures 5-7 - the same suggestions as above

5.In Table 4, as I found only strong correlation between TnI on admission and oxygen radicals (02-) followed by NT-proBNP and GSH, this results should be underlined in your results section. According to Table 2 only TNI on admission was signifcantly different between STEMI and nonSTEMI groups. Please, try to follow that results in discussion and conclusion. I suggest to reconsider the title, discussion and conclusion basing on the presented results.

Kinds

Reviewer

Author Response

(The authors gave the same response as above.)

Round 2

Reviewer 2 Report

Dear Authors,

thank you for your response.

I will keep my fingers crossed for Editor decision regarding your manuscript.

Good luck!

Reviewer